# A Method for Generating the Centerline of an Elongated Polygon on the Example of a Watercourse

**Elżbieta Lewandowicz ** and Paweł Flisek**

Department of Geoinformation and Cartography, Institute of Geodesy and Civil Engineering,
Faculty of Geoengineering, University of Warmia and Mazury in Olsztyn, 10-719 Olsztyn, Poland;
pawel.flisek@student.uwm.edu.pl
*   Correspondence: leela@uwm.edu.pl

**Abstract:** The centerlines of polygons can be generated with the use of various methods. The aim of this study was to propose an algorithm for generating the centerline of an elongated polygon based on the transformation of vector data. The proposed method involves the determination of base points denoting the direction of river flow. These points were also used to map two polygon boundaries. A Triangulated Irregular Network (TIN) was created based on the polygon's breakpoints. Edges that intersect the river channel in a direction perpendicular to river flow (across) were selected from a set of TIN edges. The polygon was partitioned into segments with the use of the selected TIN edges. The midpoints of selected TIN edges were used to generate the polygon's centerline based on topological relations. The presented methodology was tested on a polygon representing a 15-km-long section of a river intersecting the city of Olsztyn (a university center). The analyzed river is a highly meandering watercourse, and its channel is narrowed down by hydraulic structures. The river features an island and distributary channels. The generated centerline effectively fits the polygon, and, unlike the solution modeled with the Medial Axis Transformation (MAT) algorithm, it does not feature branching streams.

**Keywords:** geoinformation algorithms; centerline of a polygon; centerline of a river; watercourse modeling

---

## 1. Introduction

Inland bodies of water are represented by raster or vector data in Geographic Information System (GIS) applications. Raster data are acquired from satellite images. Datasets describing inland water bodies are generated by extracting (detecting, classifying) the information contained in images [1,2]. Vector data representing water bodies are obtained by converting classified images [3,4] or by manual vectorization of data.

Watercourses are modeled based on the data describing bodies of water. When raster data are used, the centerlines of watercourses are modeled with dedicated algorithms [5]. These tools are applied in Digital Terrain Models (DTMs), and they increase modeling accuracy by automatically generating maximum gradient lines [6]. Numerous algorithms for generating polygon centerlines based on raster data have been presented in the literature [7–10]. These solutions are based on the parallel thinning algorithm or its modified versions [11–13].

### 1.1. Generation of a Polygon Skeleton Based On Vector Data

In large-scale maps, data from classified images are converted to vector format. After data conversion, smaller rivers can be represented by lines, whereas larger rivers are represented by polygons. The resulting data are used to model watercourses. The centerlines of the modeled

watercourses are extrapolated with manual, semi-automatic or automatic methods in GIS applications. The ESRI and GRASS applications [14] for generating polygon centerlines involved semi-automatic editing options with tools for mapping midpoints. River centerlines can be generated automatically in ESRI software [14] by the "Polygon to centerline" tool.

The solutions for generating the centerlines of watercourses have been described in the literature [2,9]. Four basic methods are used for this purpose: Voronoi diagrams (VDs), the Straight Skeleton algorithm, the Medial Axis Transformation (MAT) algorithm, and triangulation (Figure 1). These methods have been presented and compared in the literature [15,16].

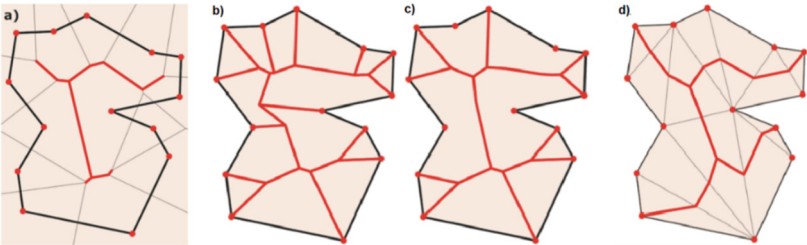

**Figure 1.** A skeleton (red lines) of the created object: (**a**) based on Voronoi diagram (VD) edges; (**b**) with the use of the Straight Skeleton method; (**c**) with the Medial Axis (Transform) method; (**d**) with constrained Delaunay triangulation [16].

In the first solution, a polygon is partitioned into segments [9,17] with the use of tools for generating Voronoi diagrams (VDs) (Figure 2).

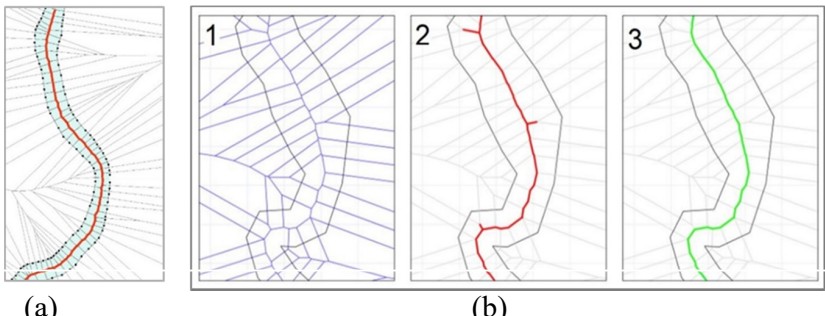

**Figure 2.** Process of generating the centerline of a river by partitioning a polygon into segments with VDs. (**a**) this study; (**b**) visualization realized by Golly and Turowski [9].

This method is deployed by various tools in programming platforms such as ET SpatialTechniques [18], GRASS GIS [19] and FME [20], as well as in Python libraries [21]. In the resulting solutions, the generated centerline covers all side streams which need to be eliminated. The generated centerlines have to be smoothed and aligned with polygon boundaries.

Polygon centerlines can also be generated with the Straight Skeleton algorithm [15,22–26], which is based on buffer lines (Figure 3). Similarly to other solutions, the Straight Skeleton algorithm first generates a skeleton of a polygon, and the centerline is then extracted from the skeleton.

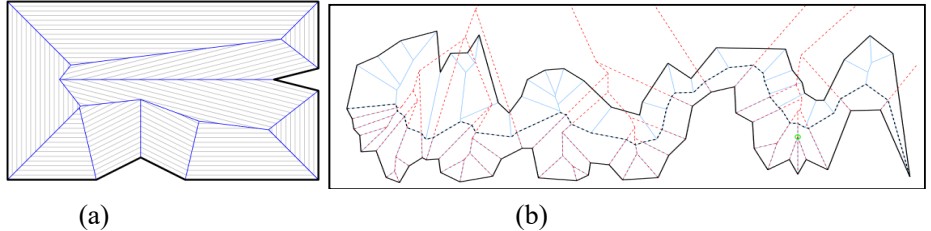

**Figure 3.** Generation of polygon centerlines with the use of the Straight Skeleton algorithm [25]; (**a**) simple example, (**b**) example at diverse polygon.

Other solutions are based on the MAT algorithm [27,28]. The centerline of a watercourse generated with the MAT algorithm features distributary channels (Figure 4) that have to be eliminated [29–33].

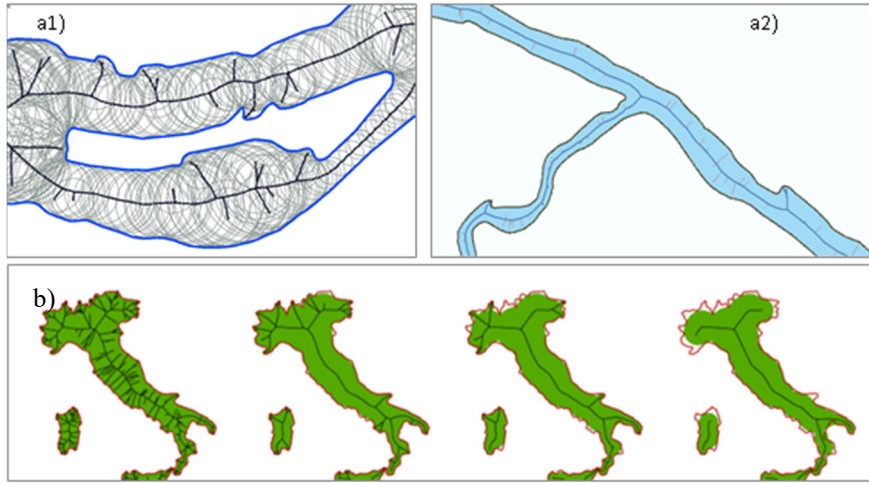

**Figure 4.** Generation of polygon centerlines with the use of the Medial Axis Transformation (MAT) algorithm: (**a1**) and (**a2**) the resulting centerline of a watercourse [34], (**b**) elimination of distributary channels during the generation of polygon centerlines on the map of Italy [35].

Delaunay triangulation has been used by many authors to generate polygon skeletons. The algorithms based on Delaunay triangulation rely on Triangulated Irregular Network (TIN) polygons, TIN edges and special locations. Three [36,37] (Figure 5(1)) or even four types of TIN polygons are identified: type 0, type 1, type 2 and type 3 [38] (Figure 5(2)), and Figure 6.

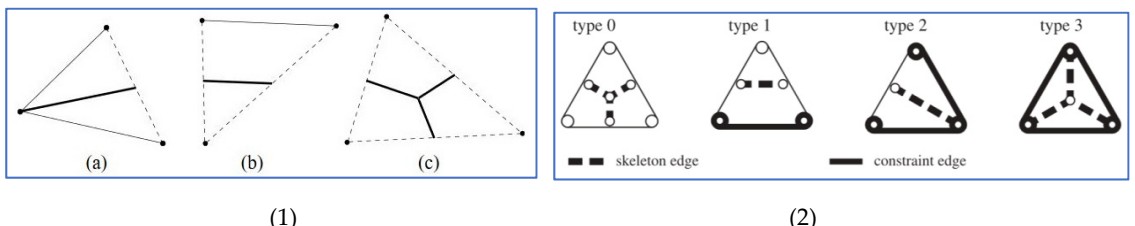

**Figure 5.** Triangle types: (**1**): (**a**) ear triangle, (**b**) link triangle, (**c**) branch triangle [37]; (**2**): 0-triangles, 1-triangles, 2-triangles and 3-triangles [38].

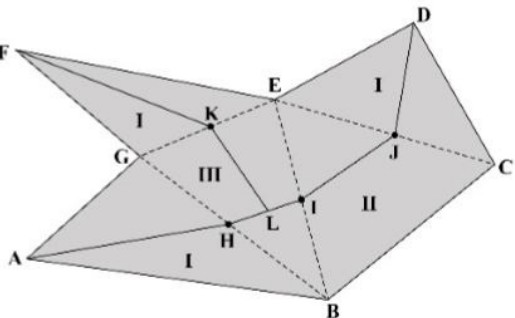

**Figure 6.** Delaunay triangle classification and the generated polygon skeleton [36].

Polygon skeletons are generated based on topological relations. The relations between special points, including midpoints of TIN edges, midpoints of TIN polygons and TIN polygons, are taken into consideration [36,37]. The polygon axis is extracted based on the generated skeleton [39–41] and the relations between the areas adjacent to the polygons. Various methods are applied for this purpose.

A split area algorithm was presented by [38] for weighted splitting of faces in the context of a planar partition which is used to generate network segments on polygons representing water bodies. The proposed solution relies on Delaunay triangulation of polygon vortices. The edges of a Triangulated Irregular Network (TIN) form a skeleton. Selected triangles are assigned weights, which facilitates correct generation of the network model (Figure 7)

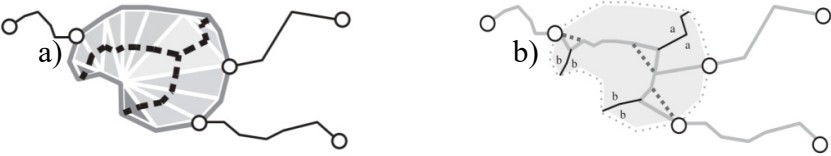

**Figure 7.** The results of a split area algorithm for generating a polygon axis based on three triangle types: (**a**) base skeleton, (**b**) modification of the final skeleton [38].

### 1.2. Methods of Generating the Axes of Elongated Polygons

Detailed solutions for generating elongated polygons based on points denoting an elongated polygon line have been proposed in the literature [40–42]. TIN edges that do not intersect the main direction of the anticipated polygon in a perpendicular direction pose a problem in the presented solutions. Side streams are created when TIN edges are taken into consideration in the process of generating a skeleton, and they have to be eliminated in successive steps. Various methods for addressing this issue have been proposed in the literature.

Wang et al. [40] used a backtracking algorithm to generate the main skeleton line by searching for other nodes in an orderly manner. This procedure involves three steps. First, a polygon is generated by Delaunay triangulation. Secondly, the start and end points of the main skeleton line are determined based on the main extension direction of the polygon. Finally, the backtracking algorithm is applied to generate the main skeleton line by searching for other nodes in an orderly manner.

Li et al. [41] proposed a partition line extraction algorithm that accounts for the direction between triangle edges and the distance of nodes in aggregation zones. Dangling arcs in the skeleton topology are eliminated by iterative deletion, and the split line is retained. The proposed method aims to maintain consistency between the characteristics of the extracted line structure and the actual structure of the object, in particular in complex branch convergence zones. In the first step, three types of aggregation patterns (aggregation zones type A, B and C) that occur in partition line extractions for LN patches of complex branch convergence zones were described using Delaunay triangulation. A partition line extraction algorithm that accounts for the direction between triangle edges and the distance of nodes in aggregation zones was then proposed.

A solution for extracting the skeleton line based on stroke features was described by [42]. In traditional approaches, the main structure and extension characteristics are difficult to maintain when dealing with dense junction areas due to the irregularity and complexity of junctions. Therefore, a skeleton line extraction method was proposed based on stroke features.

In the algorithm described by [37], TIN edges that do not intersect the polygon's main direction are transformed. The algorithm is difficult, but not impossible to compute (Figure 8). In this solution, edges were carefully flipped in the constrained Delaunay triangulation to produce high-quality axes.

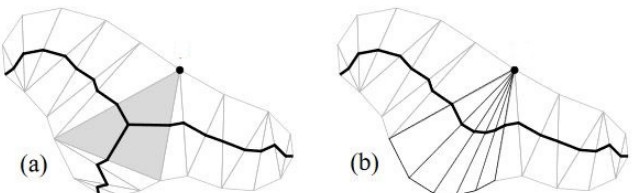

**Figure 8.** Two different axes generated from a single polygon: (**a**) an axis containing an unwanted branch, (**b**) the axis is pruned by flipping in the triangulation [37].

The described algorithms [37,40–42] for determining the main polygon axes and river axes are highly complex. New and simple solutions for automating the generation of the main river axes based on raster and vector data are thus needed. This is an important consideration because geometric models of watercourses are widely used in the management of water resources. They are also applied in specialist hydrographic research [43] and network analyses. Such solutions will also have numerous applications in cartographic generalization [44,45] and other processes [46].

### 1.3. Research Objective

The aim of this study was to propose a new and simple method for generating the centerlines of elongated polygons. It was assumed that the centerline can be generated based only on the midpoints of selected TIN edges that intersect polygons in a direction perpendicular to river flow (across). The research hypothesis states that the elimination of TIN edges that do not intersect the elongated polygon in a perpendicular direction does not exert a significant influence on the shape of the main polygon axis. Therefore, a simplified algorithm based on selected triangle types can be applied to generate the main polygon axis: (a), (b) or type 1 and type 2 triangles (Figure 5).

To verify the research hypothesis, the result produced by the simplified algorithm was compared with the centerline generated by the MAT algorithm in the ESRI application with the "Polygon to center line" tool [33] and with the centerline generated with the VDs tool. The shape of the resulting axes was compared graphically by overlapping the relevant lines and with the use of statistical data.

The proposed method was tested on vector data acquired from the public Database of Topographic Objects in 1:10,000 scale (BDOT_10) [47]. The database contains vectorized digital orthophotomaps. In the study, the river channel was represented by an elongated polygon. The BDOT_10 database features a line representing the polygon axis (river). The line was generated manually with tools for mapping midpoints, and it was compared with the axis that was generated automatically in the adopted method.

The Database of Topographic Objects in 1:10,000 scale was developed in 2010, and it sets the following requirements for editing watercourses:

- the minimum distance between line midpoints is 2 m,
- river width should be determined with an accuracy of 20%.

If the accuracy with which two river banks are determined affects the accuracy of river width and the river axis, the same approach can be applied to determine the river axis. The above requirements can be also taken into consideration in other methods for determining the river axis.

## 2. Materials and Methods

The proposed method makes a reference to the approach that had been previously deployed by [48,49] to model a network of corridors in buildings. In the present study, the described approach was used to model a river channel based on geometric data. For the needs of this study, the presented approach was modified to model elongated objects characterized by varied width and narrow segments due to the presence of hydraulic structures in the river channel. Due to variations in river width, a universal approach to selecting midpoints on polygon centerlines was adopted. In this study and in the authors' previous research, line segments intersecting polygons were generated by developing a Triangulated Irregular Network (TIN). Segments that intersect the polygon in a direction perpendicular to river flow (across) were selected by buffering (masking). The selected edges were used to generate polygon midpoints. In a watercourse with varied width, edges that intersect the river in a direction perpendicular to river flow (across) could not be selected by buffering. In the previous study, polygons were segmented with the use of VDs.

In the present study, this operation was performed based on selected TIN edges. Selected TIN edges were used to divide the polygon into segments. Segments of the polygon and the midpoints of selected TIN edges were used to generate the centerline based on topographic relations. The proposed

method is not influenced by polygon features (geometric shape), and only two points have to be determined: the beginning and the end of the river. The new method is described in detail in successive parts of the article.

The method for generating a geometric model of a polygon's centerline was tested on a section of the Łyna River. The river was represented by a polygon $\{POLYGON_{RIVER}\}$. Lines representing the river's left (Left_Bank) and right (Right_Bank) banks and the boundary lines of river islands (Island_Boundary) were derived from the polygon's boundaries. Lines denoting river boundaries in the examined area, i.e., within the administrative boundaries of the city of Olsztyn, were also derived (River Boundary_City). A set of points $\left\{POINT_{BOUNDARY}^{RIVER}\right\}$ (1) denoting breaks in polygon lines was generated based on the polygon $\{POLYGON_{RIVER}\}$.

$$\{POLYGON_{RIVER}\} \xrightarrow[generation]{} \left\{POINT_{BOUNDARY}^{RIVER}\right\}, \tag{1}$$

The resulting dataset $\left\{POINT_{BOUNDARY}^{RIVER}\right\}$ was used to generate the $\{TIN\}$, (2).

$$\left\{POINT_{BOUNDARY}^{RIVER}\right\} \xrightarrow[generation]{} \{TIN\}, \tag{2}$$

TIN edges intersecting the river channel were selected based on the polygon $\{POLYGON_{RIVER}\}$ and the generated TIN (3).

$$\{POLYGON_{RIVER}\} \cap \{TIN\} = TIN_{RIVER}, \tag{3}$$

TIN edges which touch the boundaries on opposing river banks and represent adjacent segments of the river bank were selected from the set of TIN edges $\{TIN_{RIVER}\}$ to generate the centerline of the examined river (4). This approach to selecting TIN edges is the fundamental part of the proposed methodology.

$$\{TIN_{RIVER}\} \xrightarrow[select]{} \left\{TIN_{RIVER}^{SELECT}\right\}, \tag{4}$$

In the solutions proposed by other authors, such as Meisner et al. (2016), the skeleton is generated based on the entire dataset $\{TIN_{RIVER}\}$ (3). In the next stage, the skeleton is adjusted to centerlines.

The selected edges $TIN_{RIVER}^{SELEKT}$ were used to automatically generate the midpoints of the river channel $\{MIDPOINT\}$ (5).

$$\left\{TIN_{RIVER}^{SELECT}\right\} \xrightarrow[generation]{} \{MIDPOINT\}, \tag{5}$$

The $\{MIDPOINT\}$ set contains only points inside the polygon $\{POLYGON_{RIVER}\}$. Two points denoting the beginning and end of the examined river have to be added to the dataset. These points are generated from a segment of the line denoting the boundary of the analyzed area {River Boundary_City} (6). They are the midpoints of polygon edges representing the boundaries of the examined area (administrative boundaries of the city) {River Boundary_City}.

$$\{River\ Boundary\_City\} \xrightarrow[generation]{} \{MIDPOINT_{BOUNDARY}\}, \tag{6}$$

The sum of $\{MIDPOINT\}$ and $\{MIDPOINT_{BOUNDARY}\}$ datasets is the set of midpoints of the river channel $\{POINT_{AXIS}\}$ (7).

$$\{MIDPOINT\} \cup \{MIDPOINT_{BOUNDARY}\} = \{POINT_{AXIS}\}, \tag{7}$$

The centerline of the river cannot be determined based only on dataset $\{POINT_{AXIS}\}$ alone. Neighboring points have to be identified. It was assumed that $\left\{TIN_{RIVER}^{SELECT}\right\}$ divides the polygon $\{POLYGON_{RIVER}\}\}$ into segments $\{SEGMENT_{RIVER}\}$ (8).

$$TIN_{RIVER}^{SELECT} \cap \{POLIGON_{RIVER}\} = \{SEGMENT_{RIVER}\}, \tag{8}$$

At least two points from the set $\{POINT_{AXIS}\}$ are positioned on the boundary of the polygon segment $\{SEGMENT_{RIVER}\}$. The river centerline can be generated based on the neighborhood relations between datasets $\left\{SEGMENT_{RIVER}^{base}\right\}$ and $\{POINT_{AXIS}\}$ (9). A segment of the river channel is mapped between two selected points from set $\{POINT_{AXIS}\}$ if these points are positioned on the boundary of one segment from set $\{SEGMENT_{RIVER}\}$.

$$(\{POINT_{AXIS}\}, \{\{SEGMENT_{RIVER}\}\}) \xrightarrow[generate]{} TOPOLOGY\ RIVER\ CENTERLINE \tag{9}$$

In popular solutions that rely on TIN edges, a skeleton is generated for the entire polygon based on all midpoints of TIN edges within polygon boundaries (selection of internal triangles). As a result, the polygon centerline is generated with all side streams. In the adopted methodology, data edge TIN for generating the skeleton and the centerline are prepared first. The midpoints of selected TIN edges are used to generate the polygon centerline only. Therefore, side streams are not generated in the skeleton.

The adopted methodology is illustrated by the example presented in Figure 6. The polygon contains three types of triangles. In the traditional approach, the skeleton is based on all TIN edges (Figure 5, Figure 9a). When the polygon centerline line crosses two points, for example $\overline{A-D}$, TIN edge $\overline{GE}$ does not intersect the main polygon axis. This edge should not be taken into consideration in the presented method. When this edge is eliminated, the polygon axis is determined based on selected TIN edges $\overline{(GB},\ \overline{BE},\ \overline{CE})$. The topological relations between points A, D, and the midpoints of selected TIN edges vs. three triangles and a polygon with vertices F, G, B, E are taken into account in the process of determining the polygon axis. The results are presented in Figure 9b.

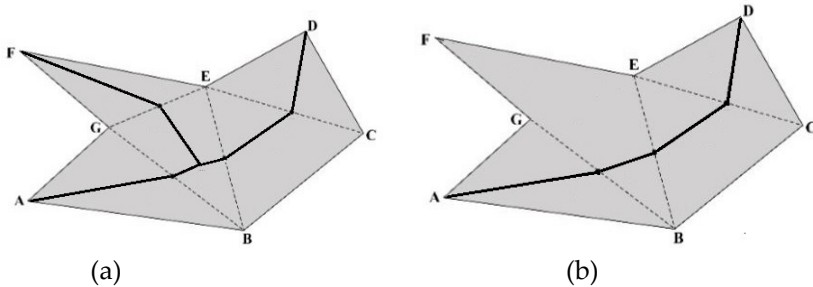

(a)          (b)

**Figure 9.** A comparison of two methods: (**a**) a traditional approach where the skeleton is based on three types of triangles; (**b**) the solution adopted in the proposed algorithm (based on the data in Figure 4).

The proposed methodology was tested on a dataset representing a fragment of river channel within the administrative boundaries of a city. The presented solution is developed with the use of GIS tools in ArcMap, ESRI [14].

## 3. Results

### 3.1. Research Site

The proposed method was tested on data from the public Database of Topographic Objects in 1:10,000 scale (BDOT_10) created in 2012. The fragment of the Łyna River within the administrative

boundaries of the city of Olsztyn in the Region of Warmia and Mazury, Poland, was represented by a single polygon (Figure 10).

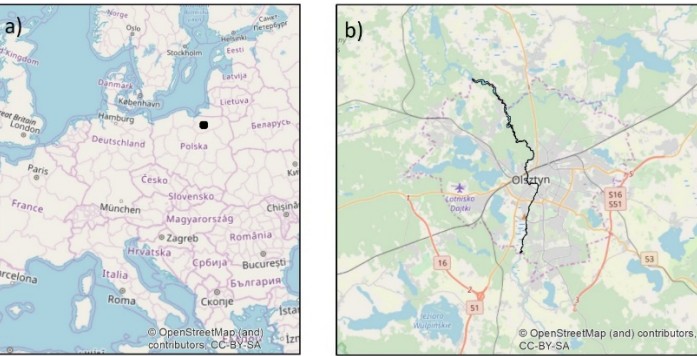

**Figure 10.** Research site in an Open Street Map (OSM): (**a**) map of Europe, (**b**) fragment of the Łyna River within the administrative boundaries of Olsztyn city.

The Łyna River is a meandering watercourse with varied channel width. The river features an island and hydraulic structures which considerably narrow down the river channel in selected locations (Figure 11). The Łyna River was selected for testing the proposed methodology due to its meandering geometry and the presence of distributary channels.

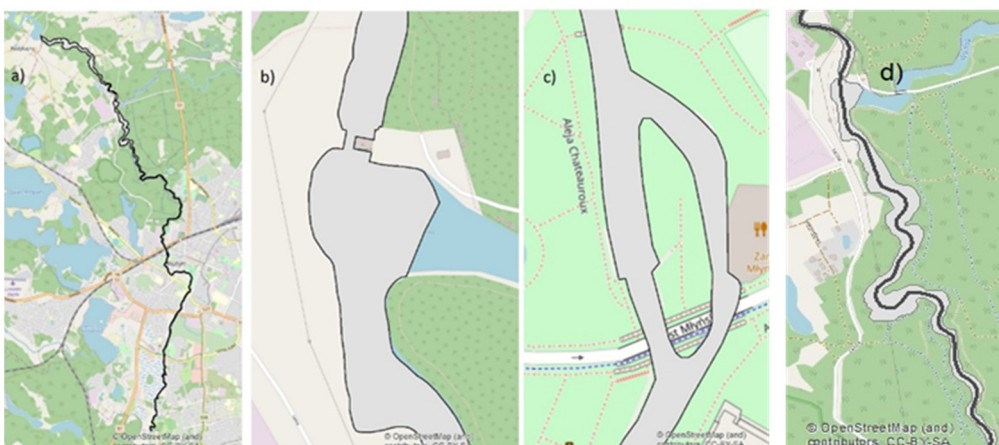

**Figure 11.** Research site: (**a**) fragment of the river within the administrative boundaries of Olsztyn; (**b**) hydraulic structure in the river channel; (**c**) river island by the hydraulic structure; (**d**) river axis from the BDOT_10 database.

The centerline of the river from the BDOT_10 database was edited semi-manually by selecting points on both river banks (polygons), and it was drawn with the software tool. The resulting centerline does not account for the river's course on both sides of the island.

## 3.2. Implementation of the Proposed Method

The edges intersecting the river channel were determined in the first stage of the study. A TIN was developed based on the points denoting river banks using the Delaunay triangulation algorithm (Figure 12, Figure 13). TIN edges positioned inside the river channel $\{TIN_{RIVER}\}$ were selected, and TIN edges touching the bank on one side of the river were eliminated. The resulting TIN edges intersect the river channel in a direction perpendicular to river flow (across) $\left\{TIN_{RIVER}^{SELECT}\right\}$ (Figure 13d, Figure 14b,d).

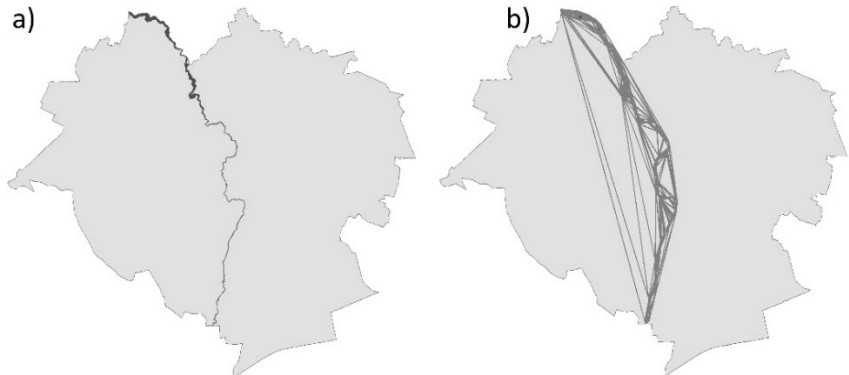

**Figure 12.** Triangulated Irregular Network Triangulated Irregular Network (TIN); (**a**) fragment of the river within the administrative boundaries of Olsztyn; (**b**) TIN model generated based on breakpoints in the river bank line within the administrative boundaries of Olsztyn city.

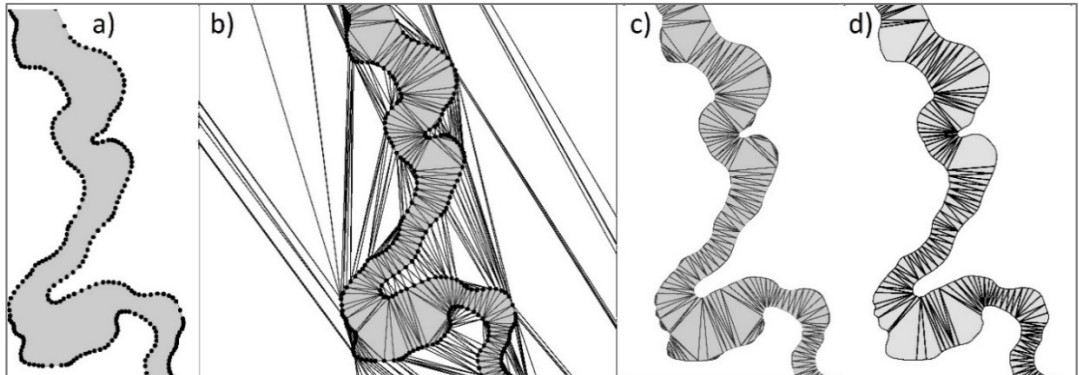

**Figure 13.** Generation of edges intersecting the river channel in a direction perpendicular to river flow (across) (**a**) points situated on the bank line, (**b**) TIN model, (**c**) TIN edges positioned inside the river channel, (**d**) selection of edges touching opposing river banks.

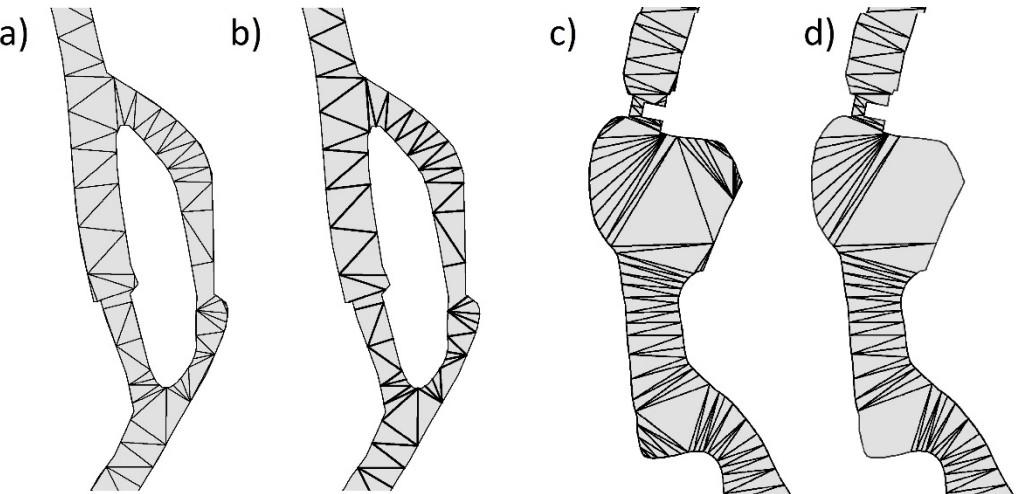

**Figure 14.** Selection of TIN edges in special locations: (**a**) island in the river channel, TIN edges in the river channel; (**b**) selected segments of TIN edges in the river channel; (**c**) hydraulic structure, TIN edges; (**d**) selected segments of TIN edges in the river channel by the hydraulic structure.

TIN edges denoting opposing river banks were selected based on topological relations. In the proposed solution, edges were selected based on the neighborhood relations between TIN edges and

the polygons adjacent to the river. Topological relations were generated based on the boundary lines of the river and the city (boundaries of the examined area), the polygon on the left bank, the polygon on the right bank, and the polygon representing the river island (Figure 15). The neighborhood relations between the river boundary and the polygons adjacent to the river are presented in Figure 15b. The "Joining Data by Location (Spatially)" tool was used to assign the ID numbers of adjacent polygons to TIN edges. TIN edges were selected with the "Select by Location" GIS tool and the "Touch the boundary of" option. These operations were iterated to analyze different neighborhood relations for all TIN edges. The TIN edges selected in successive iterations are marked in different colors in Figure 15b.

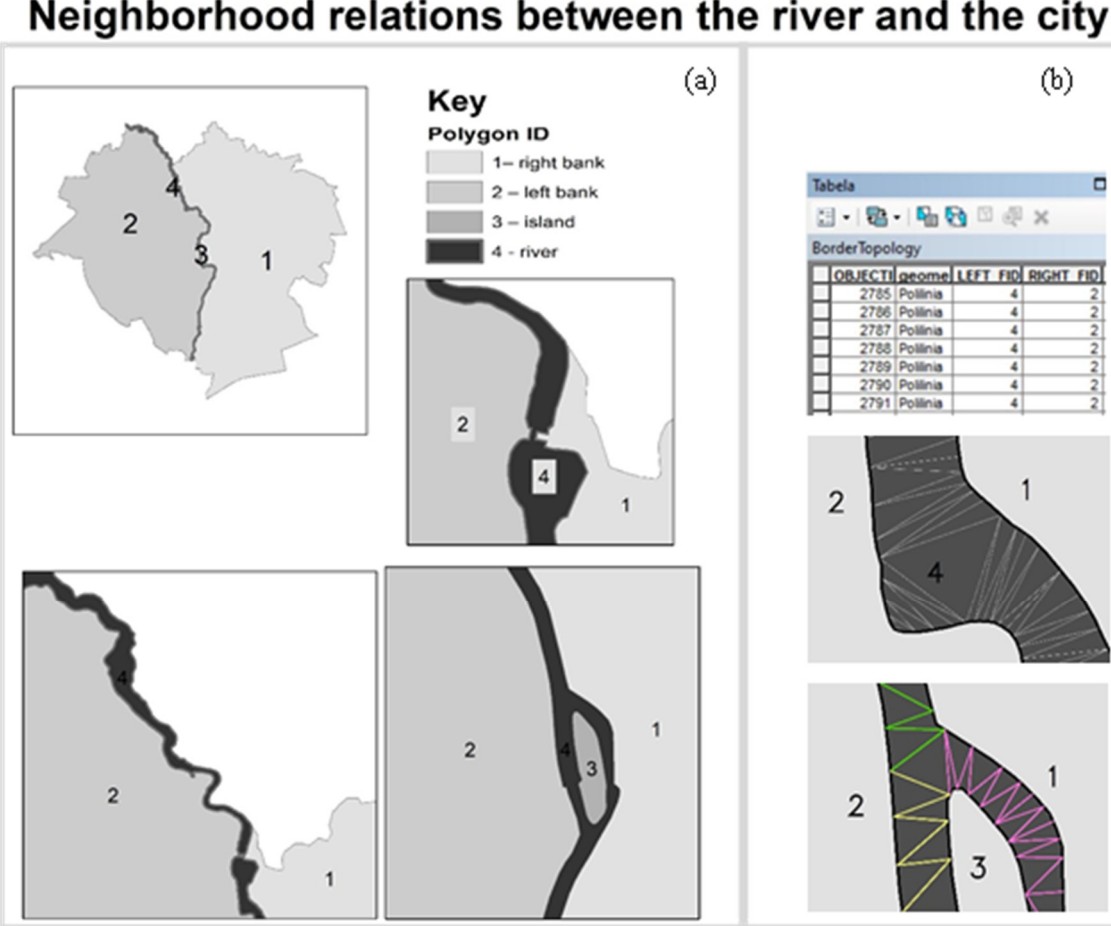

**Figure 15.** Topological (neighborhood) relations between TIN edges and polygons: (**a**) graphic presentation of neighboring polygons; (**b**) presentation of topological relations in the database, and the TIN edges selected in successive iterations based on topological relationships (marked in different colors).

The selected edges TIN $\left\{TIN_{RIVER}^{SELECT}\right\}$ intersected the river channel in a direction perpendicular to river flow (across).

The presented methodology does not generate centerlines in side streams and distributary channels. TIN edges in side streams are eliminated because they touch the polygon's boundary line on one side of the river, left or right (Figure 16).

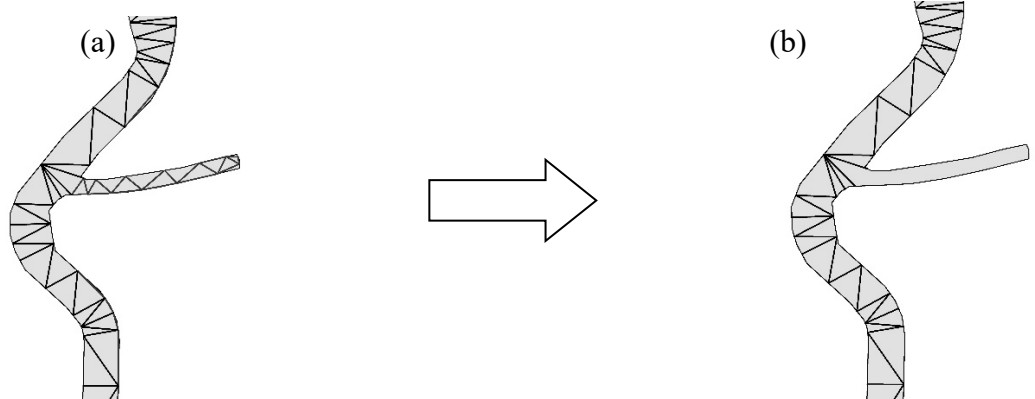

**Figure 16.** Automatic elimination of TIN channels in distributary channels during $\{TIN_{RIVER}\} \xrightarrow{select}$ $\{TIN_{RIVER}^{SELEKT}\}$ (4). The centerlines of side streams are not generated because TIN edges were eliminated. (**a**) TIN edges in the river channel $\{TIN_{RIVER}\}$; (**b**) selected TIN edged $\{TIN_{RIVER}^{SELECT}\}$.

Midpoints *{MIDPOINT} (5)* were generated based on selected edges $\{TIN_{RIVER}^{SELECT}\}$. These points are situated inside the river channel (Figure 11). The centerline of the analyzed river was generated based on the topological relations between segments $\{SEGMENT_{RIVER}\}$ (8) and points $\{POINT_{AXIS}\}$ (7). A segment of the centerline was generated between two points from the $\{POINT_{AXIS}\}$ dataset if these points touched the edge of one segment from the dataset $\{SEGMENT_{RIVER}\}$. An algorithm from the Geospatial Data Abstraction Library (GDAL) was used for data processing. The centerline generated in selected segments of the analyzed river is presented in Figure 17.

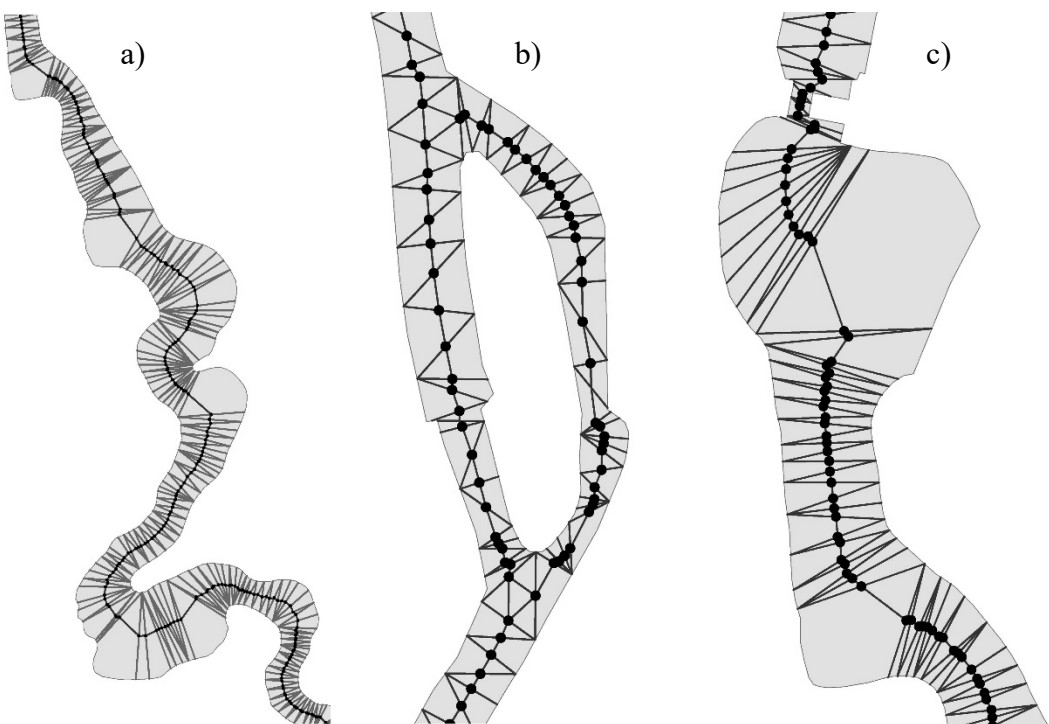

**Figure 17.** Centerline of the river channel in special locations: (**a**) floodplain; (**b**) river island; (**c**) narrow segment of the river by the hydraulic structure.

Adjustments are required in the centerline of the river segment near the island. Two segments that form loops should be eliminated. The centerline by the hydraulic structure should be adjusted.

Acute angles in the centerline should be eliminated. The centerline should be generalized with the use of the available methods [50].

### 3.3. Verification of the Proposed Method with an ESRI Tool and a Tool Based On the Voronoi Diagram

The proposed solution was verified with the use of the "Polygon to centerline" option in ESRI software that relies on the MAT algorithm. According to the description of the ESRI tool, adaptive densification is applied to the input features to improve centerline quality. The center of the polygon, generated with the ESRI tool based on the same data, represents the Łyna River, and it is shown in Figure 18 against the TIN edges from Figure 17. The network generated with the ESRI tool is largely consistent with the results presented in Figure 18, and discrepancies are observed only in the proximity of floodplains and large river bends. The centerline generated with the ESRI tool in the vicinity of the hydraulic structure runs closer to the central part of the floodplain. Downstream from the hydraulic structure, the centerline breaks at acute angles. The solution based on the TIN produced similar results. The presented method should be simplified.

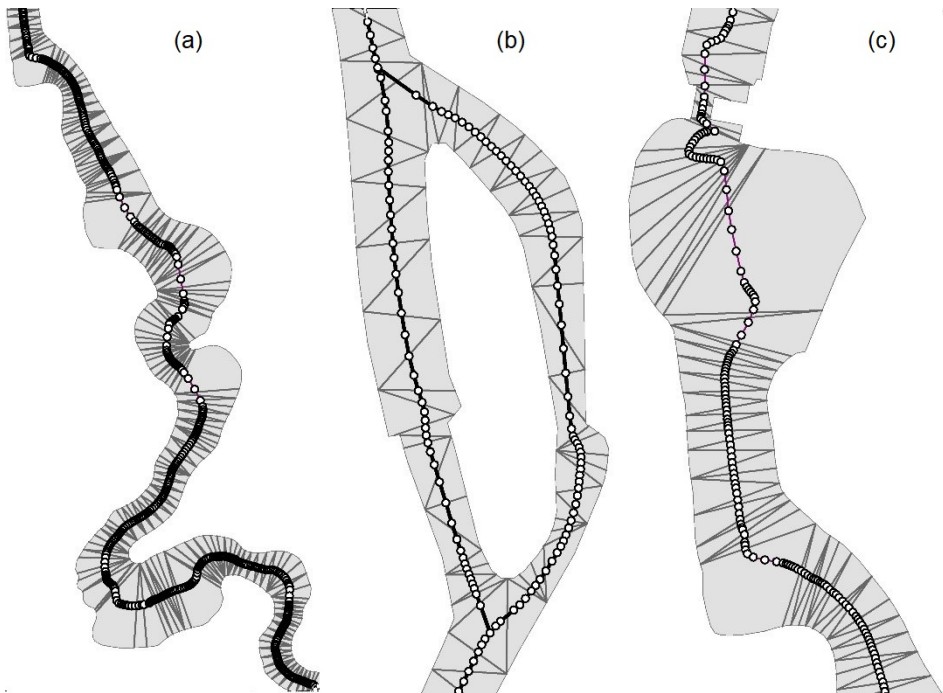

**Figure 18.** Segments of the river centerline generated with the ESRI tool and $\left\{TIN_{RIVER}^{SELECT}\right\}$ developed by the authors in special locations: (**a**) floodplain; (**b**) river island; (**c**) narrow river segment by the hydraulic structure.

According to ESRI guidelines, the input features are densified to improve the quality of the generated centerline. As a result, the centerline of the analyzed river was generated based on a much higher number of points than in the TIN approach.

The diagram in Figure 19 reveals differences between the solution developed by the authors and that generated with the ESRI tool. In the floodplain (Figure 19a), the location of the centerlines generated by the compared approaches differs by up to 15 m (the river channel has a width of around 100 m). The compared centerlines are consistent in straight-line segments of the river (Figure 19a). The solution generated with the ESRI tool is different because the input features were densified, and the output was smoothed and written as a spline.

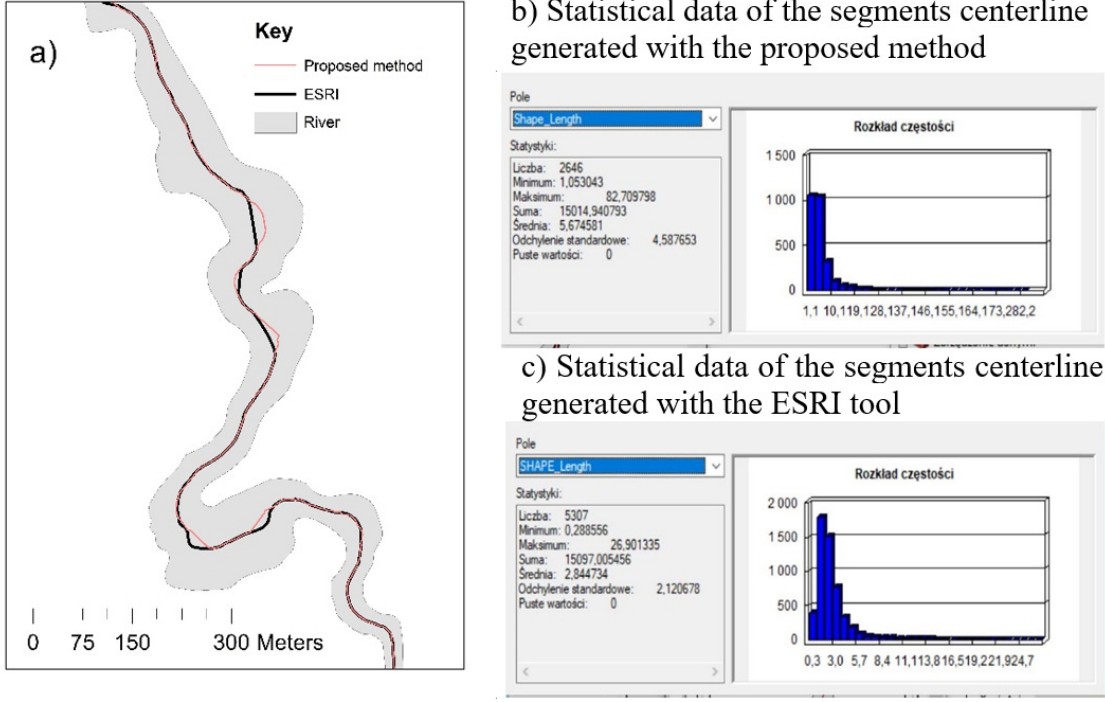

**Figure 19.** The results of the compared approaches: (**a**) graphic presentation of the compared solutions; (**b**) features of the centerline generated by the proposed method; (**c**) features of the centerline generated with the ESRI tool.

The proposed approach does not generate centerlines in distributary channels (Figure 20). The banks of side streams are incorporated into polygons representing the right or left river bank. In hydrological practice, distributary channels are usually eliminated in the process of generating a network.

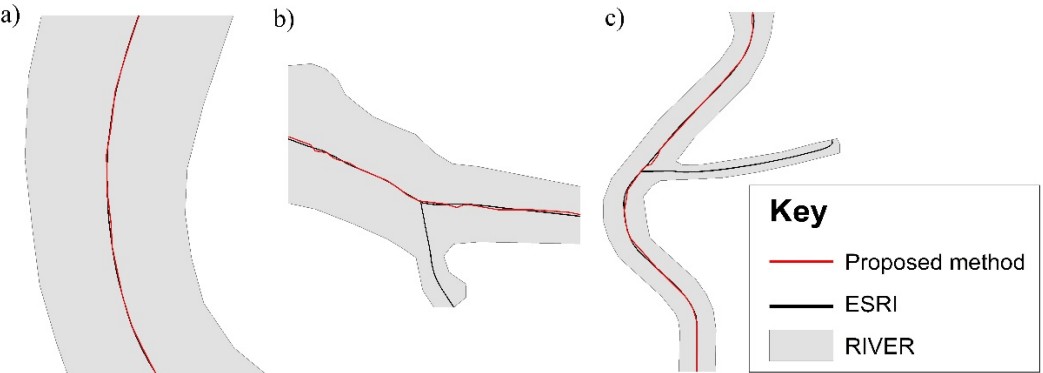

**Figure 20.** Centerlines generated by the compared methods: (**a**) centerlines are consistent in straight-line segments of the river; (**b**,**c**) the proposed solution does not generate centerlines in distributary channels.

A centerline was also generated with VDs [38]. The stages of generating a centerline are presented in Figure 15a–c. The generated skeleton had to be simplified to produce a centerline. The centerline generated in different segments of the river is shown in Figure 21d–f.

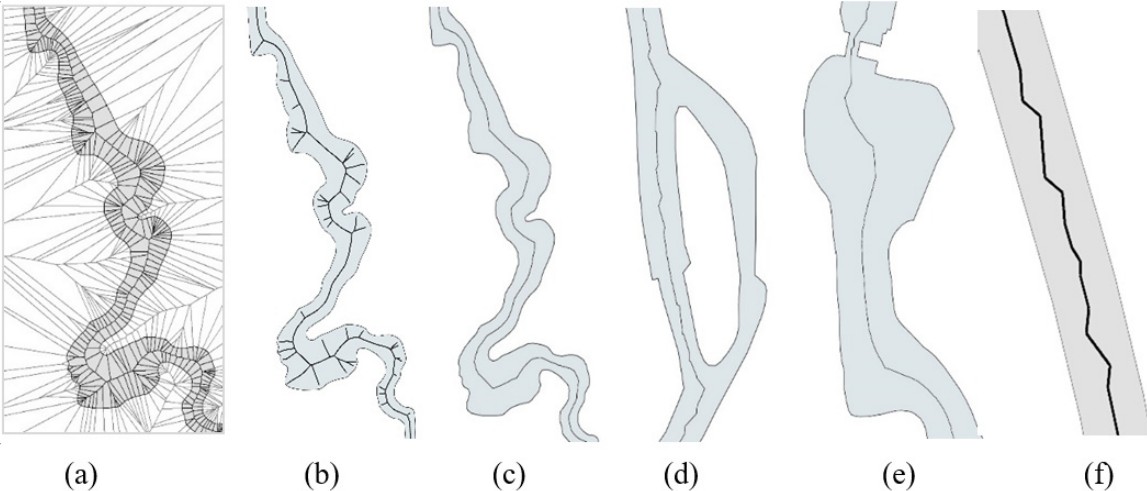

**Figure 21.** The centerline of a polygon generated with a tool based on the VD algorithm; (**a**) a view of selected river segments; (**b**) generated skeleton model. Centerlines in (**c**) the floodplain, (**b**) near the island, (**e**) by the hydraulic structure. (**f**) A zigzag centerline in straight-line segments of the river.

Additional parameters for comparing the results generated by the proposed method, the ESRI tool and the VD tool are presented in Table 1. The analyzed segment of the river has a length of approximately 15 km. The centerline generated by the proposed algorithm is 29 m shorter, and it is composed of 2646 segments (50% less than in the ESRI approach). The results were used to generate network models with the use of Network Analyst tools. The shortest paths between the beginning and end points of the analyzed river segment were generated with the use of Dijkstra's algorithm. These values can be compared because they do not contain loops and do not account for distributary channels. In the proposed solution, the Łyna River is around 26 m longer than in the model developed with the ESRI tool. The difference is less than 2% relative to river length. The river described by the centerline generated with VDs is longest, which can be attributed to the zigzag centerline in straight-line segments of the river (Figure 20f). The difference in river length determined with ESRI and VD methods is 283 m. The difference of 26 m represents only 10% of 283 m.

**Table 1.** River features in the compared approaches.

| Network Generation Method | Automatically Generated Length of the River's Center line | Number of Edges on the Centerline | Average Edge Length | Maximum Edge Length | Standard Deviation of Edge Length | River Length Described by the Shortest Path |
|---|---|---|---|---|---|---|
| Proposed method | 15,014 m | 2646 | 5.6 m | 82.7 m | 4.5 | 14,865 m |
| ESRI tool | 15,097 m The output contains distributary channels | 5307 | 2.8 m | 26.9 m | 2.1 | 14,839 m |
| Voronoi Diagram tool (VD) | 15,122 m | 3940 | 3.8 | 38.2 | 2.7 | 15,122 m |
| Manual | 14,669 m | 1048 | 13.9 | 93.6 | 12.1 | 14,669 m |

*3.4. A Comparison of Automatically and Manually Generated Centerlines*

The comparison was performed based on public resource data. Vector data describing the river channel and the centerline were obtained from the BDOT database. The river axis was generated manually. It is most highly generalized and characterized by the smallest number of segments and the longest segments. The watercourse has a length of 14,669 m. It is based on only 1048 edges, and it shorter than the shortest paths generated by the proposed method as well as ESRI and VD tools.

Three automatically generated centerlines and one manually generated centerline are presented in Figure 22. In meandering segments, the manually derived centerline better fits river bends. The centerline created with the proposed method is more consistent with the centerline generated with the VD tool and the manually generated centerline than the centerline generated with the ESRI tool. The generated centerlines are consistent in straight-line segments of the river within a margin of 1.5 m (the average width of the river channel is around 16 m), which accounts for only 9.3% of river width.

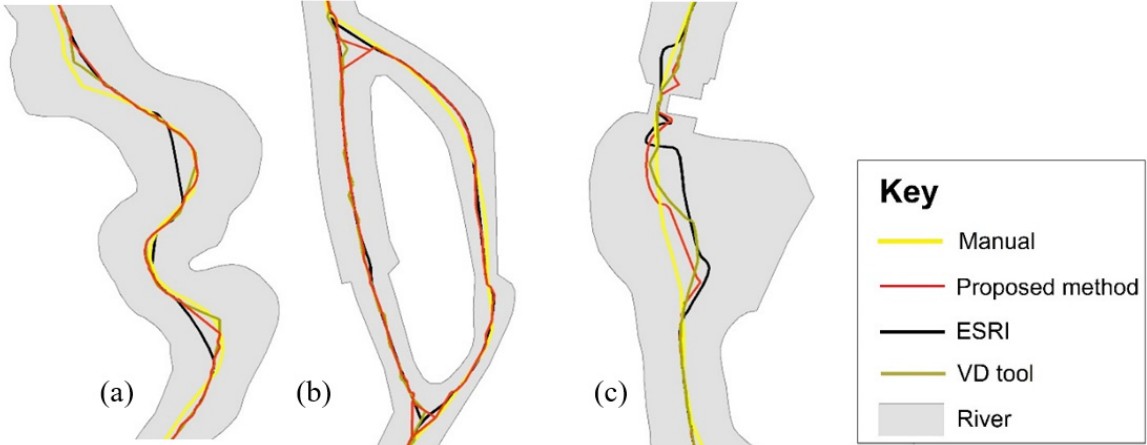

**Figure 22.** A centerline generated manually based on the image of the river channel, and the centerlines generated by the proposed method, the ESRI tool and the VD tool: (**a**) floodplain; (**b**) river island; (**c**) narrow segment of the river by the hydraulic structure.

The tools proposed by [51] were used to compare the centerlines of the river channel polygon. Four out of the 50 generated control lines intersecting the river channel are shown in Figure 23. The midpoints of control lines and the points at which all centerlines intersect control lines were generated. Four sets of river segments where the centerline deviates from the center of the river channel were created. The results are visualized in Figure 23, and the statistical parameters of the generated datasets are presented in Table 2.

The most appropriate centerline is difficult to identify. In straight-line segments of the river, the generated centerlines are similar and consistent with the average values given in Table 2. Greater deviations were noted in the floodplain and by the hydraulic structure.

All methods produced comparable results. The values generated by the manual method deviate most considerably from the values generated by the remaining methods. The polygon axis is generalized. The greatest deviation relative to river width is 14.5%. The axes developed with the use of the VD tool and the proposed algorithm are characterized by the smallest deviations. The method based on the VD tool was characterized by the lowest statistical parameters (Table 1), but the generated river was longest because its centerline is a zigzag line.

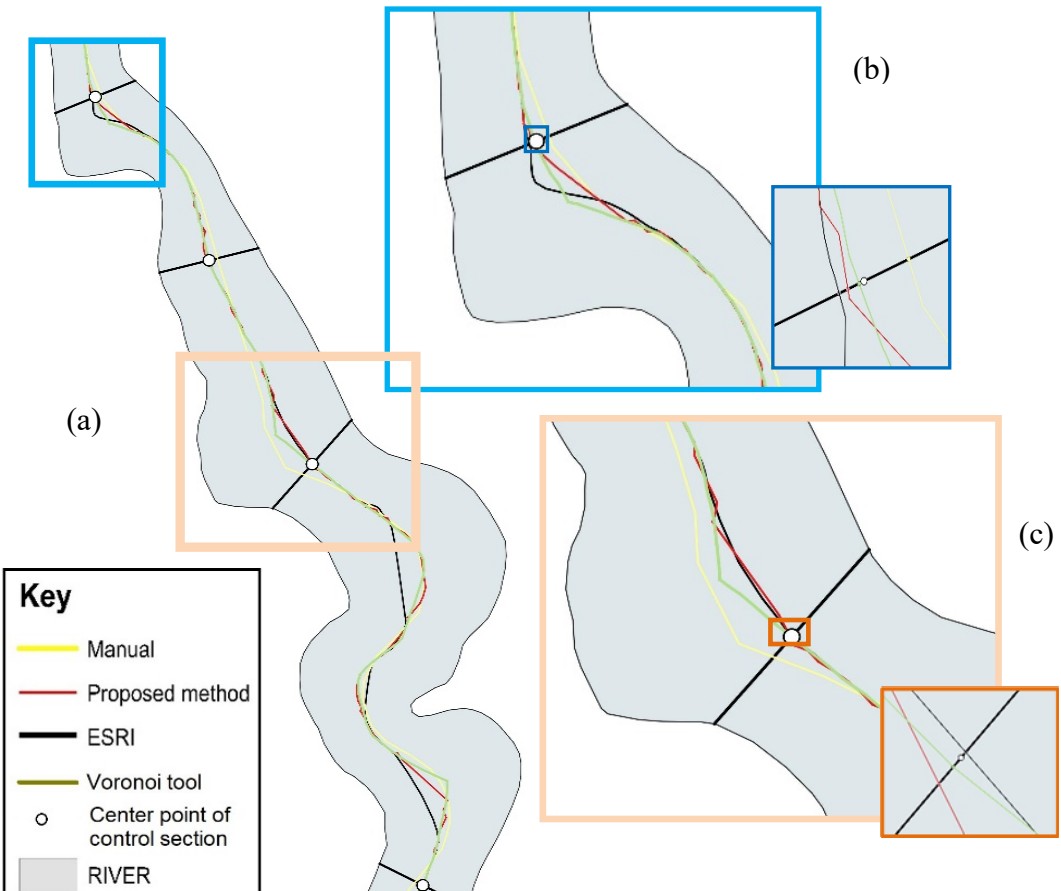

**Figure 23.** Centerlines in the control segments of the river channel; (**a**) control segments [44]; (**b**) Control line with a length of 61.68 m and the deviation of midpoints generated with different methods; (**c**) control line with a length of 86.82 m. The maximum deviation of 10.11 m was noted for the manually generated line.

**Table 2.** Statistical parameters of the centerlines in the control segments of the river channel based on the segments between the midpoints of control lines and the points where centerlines intersect control segments.

| Network Generation Method | Deviation of Central Lines from the Center of the River Channel Based on Control Segments | | | |
|---|---|---|---|---|
| | Maximum Deviation Relative to the Width of the River Channel [m]/[m] | Average Deviation [m] | Sum of 50 Deviations [m] | Standard Deviation |
| Proposed method | 1.53/61.68 | 0.24 | 12.16 | 0.29 |
| ESRI tool | 2.52/17.31 | 0.24 | 11.97 | 0.54 |
| VD tool | 0.46/61.68 | 0.13 | 6.29 | 0.11 |
| Manual | 11.66/95.19 | 2.44 | 121,76 | 2.99 |

### *3.5. A Comparison of Formal BDOT_10 Requirements and Automatically Generated Centerlines*

The provisions of the regulation setting the formal requirements regarding the precision and accuracy of the generated data should be taken into account when evaluating various methods for generating river axes. The first requirement is the distance between points, which should not be less than 2 m. The second requirement states that river width should be determined with an accuracy of up to 20%. As noted in Section 1, the accuracy of river width and the river axis depends on the

accuracy with which the river banks are determined. In the presented example, the river axis was also determined to the nearest 20%. The evaluated values are presented in Table 3.

**Table 3.** Evaluation of centerlines based on the regulation relating to the BDOT_10 database.

| Network Generation Method | Evaluation of Centerlines Based on the Regulation Relating to the BDOT_10 Database | | |
| --- | --- | --- | --- |
| | Number of Edges on the Centerline/Number Edges Shorter than 2 m | Maximum Deviation Relative to the Width of the River Channel | Accuracy of the River Axis for Maximum Deviations |
| Proposed method | 2646/140 | 1.53/61.68 | 2.5% |
| ESRI tool | 5307/2012 | 2.52/17.31 | 14.5% |
| VD tool | 3940/1070 | 0.46/61.68 | 0.7% |
| Manual | 1048/0 | 11.66/95.19 | 1.2% |

In the automatic generation method, the centerline is fragmented into a large number of segments. The highest fragmentation was observed in the ESRI method. A total of 2012 segments do not fulfill the requirements of the regulation. The smallest fragmentation was observed in the manual method, where all segments are longer than 2 m. In the proposed method, only 140 segments do not meet the relevant requirements. All automatically generated axes have to be simplified.

The accuracy of the generated axes was determined based on 50 control segments (Table 2). The axis generated by the ESRI method was characterized by the lowest accuracy, whereas the axis generated by the VD method was most accurate. It should be noted that river width was determined with an accuracy of up to 20%; therefore, the generated values are consistent with the relevant requirements.

The values generated by the proposed method require fewest modifications for consistency with the BDOT_10 requirements. The presented results testify to the effectiveness of the proposed method.

## 4. Discussion and Conclusions

The presented study proposes a new and simple algorithm for generating river centerlines. The presented solution was developed with the use of GIS tools in ArcMap.

The proposed method for generating the centerline of a river polygon is based on selected TIN edges that intersect the river channel in a direction perpendicular to river flow (across). The main river axis was generated. The centerlines of distributary channels were not generated. This approach to selecting TIN edges is a fundamental part of the proposed methodology. The described method can also be used to generate the centerline of a single river in a polygon containing a river with distributary channels.

The results of this study indicate that the proposed algorithm correctly generates the centerline of a river based on vector data. In straight-line segments of the river, the generated centerline is consistent with the centerlines generated in the remaining solutions. The centerline should be simplified in segments where the width of the river channel is significantly narrowed down by hydraulic structures. This modification is also required in the centerline generated with ESRI and VD tools.

The proposed algorithm offers an alternative to the existing solutions, and its main advantage is that it is simpler than the remaining methods. Only the main river axis is generated, without local centerlines in the river's distributary channels. The presented results indicate that the proposed method is correct and practical.

The variations in the centerlines generated based on the same set of data indicate that none of the analyzed tools are ideal. The same algorithms should be used to generate centerlines that account for variations in river geometry over time. The application of different methods to generate the centerline of a watercourse can produce erroneous results.

In the future, the proposed algorithm will be tested and modified in analyses of other objects with varied shapes, in particular a larger number of connected water bodies.

**Author Contributions:** Conceptualization, Elżbieta Lewandowicz; Methodology, Elżbieta Lewandowicz; Software, Paweł Flisek; Validation, Elżbieta Lewandowicz; Formal Analysis, Elżbieta Lewandowicz; Writing-Original Draft Preparation, Elżbieta Lewandowicz; Writing-Review & Editing, Elżbieta Lewandowicz and Paweł Flisek; Visualization, Elżbieta Lewandowicz. All authors have read and agreed to the published version of the manuscript.

**Funding:** This research was financed as part of a statutory research project of the Faculty of Geongineering of the University of Warmia and Mazury in Olsztyn, Poland, entitled "Geoinformation from the theoretical, analytical and practical perspective" (No. 28.610.033-300_timeline: 2017–2020).

**Acknowledgments:** The authors are grateful to the Regional Center for Geodetic and Cartographic Documentation in Olsztyn for providing access to the Database of Topographical Objects in 1:10 000 scale. The acquired data supported analyses of the tested objects.

**Conflicts of Interest:** The authors declare no conflict of interest.

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
