# Peer review of "A Method for Generating the Centerline of an Elongated Polygon on the Example of a Watercourse"

_ijgi, doi:10.3390/ijgi9050304_

Round 1
Reviewer 1 Report
A revision of some parts of my 1st review: "The problem is appealing and some elements of the authors methodology sound interesting. Unfortunately it seems that literature on this specific problem has been scarcely taken into account, a fact that leads in my opinion to a too modest benchmark, with respect to theory and practice:" -> literature has been fixed with some references "the goals are not well defined and hence the comparison misses criteria for objective and (possibly) numerical evaluation." -> goals and comparison is still lacking some more robust argument, e.g.: cite some work in the literature assessing the need for a method discarding distributary channels in order to sustain one of the claimed advantages of the methodology; how and by whom the manually generated centerline was made? can it be taken as a sort of ground truth?; -> criteria for evaluation is missing, even if a new table with some numerical evaluation has been added: it is not explained how the numbers of the new table should support an evaluation of the proposed methodology (how, for example, the average edge length of the centerlines resulting from the applied methods should contribute to their evaluation?). The computational complexity of the proposed methodology has not been evaluated nor compared with the alternative algorithms. I appreciated the computation of some more statistical parameters (table2) but the authors should design a theoretical framework to interpret the results. -> Even the bibliography, as said, has been appropriately expanded, the following statement from the first round of my review still persists, especially for what concerns absence of evaluation criteria for the claimed “validation”: "It is difficult for the authors to demonstrate the novelty and the effectiveness of their proposal in the absence of a more accurate bibliographic study and a comparison with alternative methodologies, to say nothing of the complete absence of convincing evaluation criteria (the claimed "validation" of the methodology unfortunately appears really not much more than an aesthetic comparison of the results)." -> I would have appreciated some authors’ responses to reviewers' comments (not only mine), but is seems that the authors limited themselves to add paragraphs and some material to their work, a fact that, in particular considering 2 “reject” over 4 reviewers, seems quite inappropriate. -> As I said in the first round, the methodology seems appealing, but the weak research and the evaluation framework (just one limited case study, no criteria definition) make me say the work is not yet ready for publication. If it was the first round I would suggest a further revision to offer the authors the possibility to answer to remaining issues.Author Response
Please see the attachment.

Reviewer 2 Report
- What is (a) and (b) in Figure 1 and (b) in Figure 2?
- Formula 4 and other formulas, SELEKT→SELECT?
- L195, Poligon→Polygon
- L395, Pyton→Python
Author Response
Thank you for the time and effort taken to review the revised manuscript. All comments and suggestions have been taken into account in this round of the revision process.
Reviewer 3 Report
Accept in present form.
Author Response
Thank you for the time and effort taken to review the revised manuscript.
Round 2
Reviewer 1 Report
The paper have been greatly ameliorated, bibliography has been properly expanded and the research question and the validation framework are now clearly stated, letting the results to be more clear.
I consider the paper ready for publication.
Just check for minor typos (e.g. line 99 the typo "vOrtices"... I guess "vertices" was intended).
Well done.
This manuscript is a resubmission of an earlier submission. The following is a list of the peer review reports and author responses from that submission.
Round 1
Reviewer 1 Report
The paper describes a method, or better, a pipeline involving different existing GIS tools in (ESRI) ARCMAP commercial software. The pipeline leads to the obtainment of an elongated polygon centerline approximation. The authors apply the method to the case of rivers, namely to the specific case of a segment of the Lyna River. The output of this very specific case study is compared to the output of a similar tool offered by the same ESRI software, which involves the Medial Axis Transformation algorithm.
The authors highlight two advantages of their proposal with respect to the ESRI tool: 1) the result of the proposal does not feature distributary channels 2) the result better fits a "manual" delineation of river centerline.
The problem is appealing and some elements of the authors methodology sound interesting. Unfortunately it seems that literature on this specific problem has been scarcely taken into account, a fact that leads in my opinion to a too modest benchmark, with respect to theory and practice: the goals are not well defined and hence the comparison misses criteria for objective and (possibly) numerical evaluation. With this respect the presented state of the art focuses and cites many works about algorithm concerning thinning (an algorithmic task with a completely different domain - raster images) and a very scarce sample of works about polygon centerlines/skeleton extraction methods from vector data: in the introduction, that the author devote to the state of the art, the main citations of works about the specific problem they intend to consider (extracting skeleton line of an elongated polygon) are ESRI software online documentation, a forum webpage with some discussion answering the question about "how can I compute the centre line of road polygon?".
As a consequence, the rest of the study is in my opinion too poorly designed being limited to the presentation of the methodology (quite interesting), and the comparison with just the ESRI tool. It is difficult for the authors to demonstrate the novelty and the effectiveness of their proposal in the absence of a more accurate bibliographic study and a comparison with alternative methodologies, to say nothing of the complete absence of convincing evaluation criteria (the claimed "validation" of the methodology unfortunately appears really not much more than an aesthetic comparison of the results).
Reviewer 3 Report
Extracting centerlines from dual-line or polygon is very important in hydrology modeling and spatial analysis. The present paper proposed an algorithm for generating the centerline of an elongated polygon based on TIN edges and topological relations. However, this paper will need substantial improvements to make it a truly outstanding contribution to the literature.
1.There are lots of methods and tools to extract centerline besides of ArcGIS or MAT. It is included the v.centerline function in GRASS and related tools in ET_GeoWizards or FME, etc. The paper should make a detailed comparison between the proposed method and the existing methods or tools mentioned above, to highlight the advantages of the proposed method.
2.The proposed algorithm is not described in detail in Section 3. Algorithm flowchart or code or pseudo code should be provided to verify the rationality and reproducibility of the method. At the same time, the proposed algorithm is similar to the centerline package using Python language, and further comparison and analysis should be discussed.
3.The experiment is not substantial enough too. It lacks multiple test cases with different shapes and morphologies, as well as complete water bodies including main and tributary streams; it is impossible to judge the applicability of the algorithm. There is no test case for large volumes of data (such as multiple rivers, longer rivers), and the performance of the algorithm cannot be evaluated.
4.The paper structure needs to be further considered. Section 2 should be incorporated into the Introduction section, and the Related works section should be added to the paper.
Compared with the existing methods and algorithms, the advantages of the proposed method are not outstanding. Although the paper proposes that the algorithm can eliminate distributary channels, it lacks an in-depth analysis of the causes for this feature. There are a series of defects in algorithm description and testing. In summary, the paper has not yet met the IJGI publication requirements, and it is recommended to reject the manuscript.
Reviewer 4 Report
Dear authors,
Your paper elaborates the issue of ''A method for generating the centerline of an elongated polygon on the example of a watercourse''. The paper elaborates very good previous research done in this matter. In my opinion methodology is appropriate and results are well documented. All the sections are well described in a formal yet understandable approach. It is clear that the research method and results are scientifically presented.
English is at an academic level.
Congratulations to the authors on their very interesting research
Here are some minor corrections and suggestions:
In the pdf version of the document I received the figures that are not of required quality, so this needs to be improved. The details in some figures are very poor and vague (especially in Figure 8 and Figure 11). If there is no limit on the number of pages, it would be good to see the results of compared approaches (as it was given in Figure 11) for the case of river island and the case of narrow river segment by the hydraulic structure.